# Cocoa Extract Provides Protection against 6-OHDA Toxicity in SH-SY5Y Dopaminergic Neurons by Targeting PERK

**DOI:** 10.3390/biomedicines10082009

**Published:** 2022-08-18

**Authors:** Vincenzo Vestuto, Giuseppina Amodio, Giacomo Pepe, Manuela Giovanna Basilicata, Raffaella Belvedere, Enza Napolitano, Daniela Guarnieri, Valentina Pagliara, Simona Paladino, Manuela Rodriquez, Alessia Bertamino, Pietro Campiglia, Paolo Remondelli, Ornella Moltedo

**Affiliations:** 1Department of Pharmacy, University of Salerno, I-84034 Fisciano-Salerno, Italy; 2Department of Medicine, Surgery and Dentistry “Scuola Medica Salernitana”, University of Salerno, I-84034 Baronissi-Salerno, Italy; 3Department of Chemistry and Biology, I-84034 Fisciano-Salerno, Italy; 4Department of Molecular Medicine and Medical Biotechnology, University of Naples Federico II, I-80131 Naples, Italy

**Keywords:** Parkinson’s disease, unfolded protein response, endoplasmic reticulum stress, PERK, oxidative stress, cocoa

## Abstract

Parkinson’s disease (PD) represents one of the most common neurodegenerative disorders, characterized by a dopamine (DA) deficiency in striatal synapses and misfolded toxic α-synuclein aggregates with concomitant cytotoxicity. In this regard, the misfolded proteins accumulation in neurodegenerative disorders induces a remarkable perturbations of endoplasmic reticulum (ER) homeostasis leading to persistent ER stress, which in turn, effects protein synthesis, modification, and folding quality control. A large body of evidence suggests that natural products target the ER stress signaling pathway, exerting a potential action in cancers, diabetes, cardiovascular and neurodegenerative diseases. This study aims to assess the neuroprotective effect of cocoa extract and its purified fractions against a cellular model of Parkinson’s disease represented by 6-hydroxydopamine (6-OHDA)-induced SH-SY5Y human neuroblastoma. Our findings demonstrate, for the first time, the ability of cocoa to specifically targets PERK sensor, with significant antioxidant and antiapoptotic activities as both crude and fractioning extracts. In addition, cocoa also showed antiapoptotic properties in 3D cell model and a notable ability to inhibit the accumulation of α-synuclein in 6-OHDA-induced cells. Overall, these results indicate that cocoa exerts neuroprotective effects suggesting a novel possible strategy to prevent or, at least, mitigate neurodegenerative disorders, such as PD.

## 1. Introduction

The endoplasmic reticulum (ER) is a vast network of membranes engaged in a variety of important functions such as Ca^2+^ signaling [1], control of protein folding and vesicular transport of cargo proteins [2,3], and lipid and carbohydrate metabolism [4,5]. ER functions are impaired once the organelle undergoes stress, normally referred to as ER stress, which is generated by the accumulation of misfolded protein aggregates as well as the overload of viral proteins, altered Ca^2+^ balance or metabolic dysfunctions [6,7]. To counteract misfolding within the ER, eukaryotic cells react by activating an adaptive signalling response, known as unfolded protein response (UPR) [8] mostly controlled by three ER-transmembrane proteins, which, by different mechanisms, trigger the UPR: the (PKR)-like ER kinase Ser/Thr protein kinase (PERK), the inositol-requiring Ser/Thr protein kinase 1α and RNA endonuclease (IRE1α) and the activating transcription factor 6 (ATF6) [9,10]. Each transducer activates a distinct pathway aiming to enhance ER protein-folding capacity, to reduce ER protein load, to modulate the ER exit of newly synthesized proteins and to potentiate ER-associated degradation (ERAD) or autophagy [11,12]. Therefore, the UPR pathways normally overcome stress and restore the ER proteostasis and function. On the other hand, when ER stress overwhelms the UPR and the ER proteostasis is not quickly fixed, the pro-survival mission of the UPR turns out into a suicidal response [13,14]. This outcome is frequent in various pathological conditions such as cancer and metabolic diseases and is particularly common in neurodegenerative disorders [15,16,17,18,19,20].

Parkinson’s disease (PD) is a widespread neurodegenerative pathology affecting about 2–3% of the over-65s individuals. PD arises from genetic or environmental reasons, which cause death of dopaminergic neurons in the substantia nigra pars compacta with subsequent deficiency of the dopamine neurotransmitter [21,22,23,24]. Beyond that, at cellular level mitochondrial disfunction, alterations of reactive oxygen species (ROS) homeostasis [25,26,27,28,29,30,31,32] and unsuccessful protein folding control together with impaired autophagy [33,34,35] are the main features of PD. Since ER protein folding is coupled to the ROS release, oxidative stress (OS) can be triggered by ER stress due to the more intense folding work inducing a higher production of ROS [36,37].

Neuronal cell death in PD is often associated to the activation of the PERK pathway [38,39,40]. For this reason, either the pharmacological blockage of PERK, by using inhibitors such as GSK2606414, or PERK gene knock-down strategies prevent the neurodegeneration in experimental models of PD [41,42].

Cocoa and cocoa-derived products contain several substances, such as polyphenols or flavanols such as epicatechin and catechin, widely known for their excellent antioxidant activity [43]. Generally, these compounds improve neuronal protection enhancing endothelial cells NO synthesis and, thus cerebrovascular circulation [44,45,46]. In addition, some cocoa-derived products can easily cross the blood-brain barrier [47] exerting the antioxidant activity directly in neuronal cells by regulating specific signaling pathways [44,48,49].

Here, we investigate if cocoa extracts could preserve SH-SY5Y dopaminergic neurons from 6-OHDA cytotoxicity correlating these results to ER stress inhibition. In details, we noticed that a mix of cocoa procyanidins was able to block 6-OHDA toxicity inhibiting ER stress. Remarkably, we were able to correlate the protective mechanism of the cocoa extract to the selective inhibition of PERK autophosphorylation, which in turns results in the protection of SH-SY5Y dopaminergic neurons from OS, α-synuclein accumulation and apoptosis ER stress-mediated.

## 2. Materials and Methods

### 2.1. Sample Preparation

*Theobroma cacao* L. unfermented and unroasted beans were manually crushed into fine powder. After 24 h, the extract obtained by ethanolic extraction from maceration was centrifuged and then the supernatant was collected, evaporated to dryness under vacuum at 30 °C in a rotary evaporator, and finally dissolved in water, filtered through a 0.45 μm nylon filter membranes and then lyophilized.

### 2.2. RP-UHPLC-PDA-ESI-MS/MS Analysis

The analytical characterization of phytochemical compounds contained in the cocoa extract was performed on a Shimadzu Nexera RP-UHPLC-PDA system (Reversed Phase-Ultra High Performance Liquid Chromatography-Photodiode Array, Kyoto, Japan) coupled online with a LC-MS-IT-TOF (Liquid Chromatography-Mass Spectrometry-Ion Trap-Time of Flight, Shimadzu, Milan, Italy) equipped with an electrospray source (ESI). LC-MS data elaboration was performed by the LCMS solution^®^ software (Version 3.50.346, Shimadzu, Milan, Italy).

Chromatographic separation was accomplished on a Kinetex^®^ EVO C18 column (150 × 2.1 mm × 2.6 µm, 100 Å, Phenomenex^®^, Bologna, Italy) maintained at 45 °C, employing H_2_O and ACN plus 0.1% (*v*/*v*) CH_3_COOH as mobile phases delivered at constant flow rate of 0.5 mL min^−1^. Analysis was performed in gradient elution as follows: 0–10 min, 5–26% ACN; 10–13 min, 26–95% ACN; isocratic at 95% for 3 min and, finally 5 min for column re-equilibration. Data acquisition was set in the range 190–800 nm and chromatograms were monitored at 280 nm.

MS detection of bioactive compounds was operated in positive and negative mode ionization. Full scan MS data were acquired in the range 150–1500 *m/z* and MS/MS experiments were conducted in data dependent acquisition. Molecular formulas of identified compounds were determined by “Formula Predictor” software (Shimadzu, Milan, Italy).

### 2.3. Semiprep-RPHPLC-UV/Vis

The purification of bioactive compound identified in the cocoa extract was carried out by semi-preparative reversed phase liquid chromatography employing a Shimadzu Semiprep-HPLC system consisting of two LC-20AP pumps, a LH-40 autosampler, a UV detector SPD-40V equipped with a preparative cell and a system controller CBM-40. The separation was carried out on a Luna^®^Omega C18 column (250 × 10 mm × 5 μm, 100 Å), employing as mobile phases water (A) and acetonitrile (B) both acidified by 0.1% (*v*/*v*) CH_3_COOH setting the flow rate at 5 mL min^−1^. The analysis was performed in gradient elution as follows: 0–15 min, 5–30% B; 15–18 min, 30–95% B; 18–22 min, isocratic to 95% B, then five minutes for column re-equilibration. The cocoa extract was purified in three aliquots: fraction I containing theobromine, fraction II constituted by catechin-3-*O*-glucoside and N-caffeoyl-L-aspartate and, fraction III composed by epicatechin and procyanidins.

### 2.4. Cell Culture and Drug Treatment

The human neuroblastoma SH-SY5Y cell line was obtained from American Type Culture Collection (ATCC, Rockville, MD, USA). Cells were grown at 37 °C in a 5% CO_2_ atmosphere in Dulbecco’s Modified Eagle Medium (DMEM, 4500 mg/mL glucose) supplemented with 10% (*v*/*v*) fetal bovine serum, 2 mM L-glutamine, 100 U/mL penicillin, and 0.1 mg/mL streptomycin. In each experiment, cells were placed in a fresh medium, cultured in the presence of the cocoa raw extract or its fractions, and followed by adding 6-OHDA (Sigma Aldrich, St. Louis, MO, USA) EC_50_.

EC_50_ values were calculated using GraphPad Prism 8.0 software (San Diego, CA, USA) by nonlinear regression of dose-response inhibition.

### 2.5. Cell Viability Assay

Cell viability was established by measuring mitochondrial metabolic activity with PrestoBlueTM (PB) (Cat. N. A13262, Invitrogen), employed according to the manufacturer’s protocol [50]. In brief, SH-SY5Y (5 × 10^4^ cells/well) were plated into 96-well plates for 24 h, then the cocoa raw extract (1–50 µg/mL) or its fractions (1–6 µg/mL) were added for 1 h. Next, 6-OHDA was added for 24 h. Afterward, PB reagent at 10% final concentration for 2 h was added. The absorbance was measured at 570 nm, with a reference wavelength set a 600 nm, and using a microplate reader (Multiskan Go, Thermo Scientific, Waltham, MA, USA). Cell viability was expressed as a percentage relative to the untreated cells cultured in medium with 0.1% DMSO and set to 100%, whereas 10% DMSO was used as positive control and set to 0% of viability.

### 2.6. Determination of Hypodiploid Nuclei

Hypodiploid nuclei were analyzed using propidium iodide staining by flow cytometry as described previously [51]. In more detail, SH-SY5Y cells (4 × 10^5^ cells/well) were grown in 12-well plates and allowed to adhere for 24 h. Later the medium was replaced, and cells were pretreated with fraction III (6 µg/mL) for 1 h and then exposed to 50 µM 6-OHDA for another 24 h. After treatment, the culture medium was replaced, cells washed twice with PBS and then suspended in 500 μL of a solution containing 50 µg/mL propidium iodide, 0.1% (*w*/*v*) sodium citrate, and 0.1% Triton X-100. Culture medium and PBS were centrifuged, and cell pellets were pooled with cell suspension to retain both dead and living cells for analysis. After incubation at 4 °C for 20 min in the dark, cell nuclei were analyzed with a Becton Dickinson FACScan flow cytometer using the Cell Quest software, version 4 (Franklin Lakes, NJ, USA). Cellular debris was excluded from the analysis by raising the forward scatter threshold, then the percentage of cells in the hypodiploid region (sub G0/G1) was calculated.

### 2.7. Colony Formation Assay

The clonogenic potential [52] was assessed using a subtoxic dose (20 μM) of the 6-OHDA and fraction III 6 µg/mL. Cells were plated in 6-well plates at a seeding density of 2 × 10^3^ cells/well. After incubation for 10 days, the culture was terminated by removing the medium and washing the colonies twice with PBS. The cells were fixed and stained with a solution containing 3.7% formaldehyde and 0.5% crystal violet for 30 min, and then washed twice with PBS. Images were obtained and the number of colonies was counted with the free image-processing software ImageJ, version 1.47 (http://rsb.info.nih.gov/ij/, accessed on 1 February 2022).

### 2.8. 3D Cell Culture Generation

Spheroids were obtained slightly modifying the protocol previously reported [53]. In more details, 20 μL drops of the culture medium solution containing 4 × 10^3^ cells were pipetted onto the lid of 100 mm culture dishes and were inverted over dishes containing 10 mL of PBS. Hanging drop cultures were incubated for 4 days, to allow sedimentation. The resultant cell aggregates were harvested, and each spheroid was gently transferred into agarose-coated 12-well plate for 10 days treatments with 6-OHDA (20 μM) and fraction III (6 µg/mL). Images were captured using LEICA ICC50 HD light microscope (10×) (Leica Microsystems, Wetzlar, Germany).

### 2.9. ROS and NO Detection

Reactive oxygen species (ROS) and nitric oxide (NO) levels were measured using 10 μM 6-carboxy-2′,7′-dichlorodihydrofluorescein diacetate (DCFH-DA, Sigma Aldrich, St. Louis, MO, USA) and 5 μM 4-amino-5-methylamino-2′,7′-difluorofluorescein diacetate (DAF-FM Diacetate, Thermo Fisher Scientific, Waltham, MA, USA). To test the effect of fraction III (6 µg/mL) to ROS and NO neutralization, SH-SY5Y cells were seeded (3 × 10^4^ cells/well) in 12-well plates allowing to adhere for 24 h. Next, cells were pre-incubated for 1 h with fraction III (6 µg/mL) using 6-OHDA 50 μM as stressor. After 24 h, the medium was removed, and the cells were washed twice with PBS. A staining solution containing DCFH-DA or DAF-FM in serum-free medium without phenol-red was added for 15 min at 37 °C in the dark. The fluorescence signals were evaluated using Becton Dickinson FACScan flow cytometer and analyzed with Cell Quest software, version 4 (Franklin Lakes, NJ, USA).

### 2.10. Western Blotting Analysis

SH-SY5Y cell line was seeded in 60 mm culture dishes, treated alone with fraction III (6 μg/mL), 6-OHDA (50 μM) or with their combination. Thapsigargine was used as positive control of ER stress induction, while GSK2606414 was used as inhibitor of PERK phosphorylation. After 4 h, cells were washed twice with PBS and detached with a scraper, centrifuged for 5 min at 655× *g* at 4 °C. Full proteins were extracted by lysis buffer (20 mM Tris-HCl pH 7.5, 150 mM NaCl, 1 mM Na_2_EDTA, 1 mM EGTA, 2% NP-40, 1% sodium deoxycholate, 1× protease and phosphatase inhibitor cocktail) for 30 min. Thereafter, cell lysates were centrifuged at 4850× *g* for 20 min at 4 °C. 30 μg of total proteins were run on 8–10% SDS-PAGE and transferred to nitrocellulose membranes using a minigel apparatus (Bio-Rad Laboratories, Hercules, CA, USA). Blots were blocked in phosphate buffered saline, containing Tween-20 0.1% and 10% nonfat dry milk for 2 h at room temperature and incubated overnight with specific primary antibodies at 4 °C with slight agitation. GAPDH and α-tubulin were used as the loading control. The following antibodies were used: rabbit monoclonal anti-PERK (Cell Signaling, Danvers, MA, USA), rabbit polyclonal anti-phospho-eif2α (Cell Signaling, Danvers, MA, USA), mouse monoclonalanti-ATF6 (Cell Signaling, Danvers, MA, USA), anti-caspase-12 mouse monoclonal, anti-GAPDH (Santa Cruz Biotechnology, Dallas, TX, USA), mouse monoclonal anti-α-tubulin (Santa Cruz Biotechnology, Dallas, TX, USA). After washes in PBS/Tween-20 0.1%, the appropriate anti-rabbit or anti-mouse (Pierce, Thermo Fisher Scientific, Waltham, MA, USA) peroxidase-linked secondary antibody was added for 1 h at room temperature. Antigen-antibody complexes were detected by enhanced chemiluminescence (ECL kit, Amersham, Germany). Filters were exposed to LAS 4000 (GE Healthcare, Chicago, IL, USA) and the densitometry analysis of autoradiographs was performed by the ImageJ program, version 1.47 (http://rsb.info.nih.gov/ij/, accessed on 1 February 2022) [54].

### 2.11. RT-PCR and XBP1 Splicing Assay

SH-SY5Y cells were seeded in 100 mm culture dishes and treated alone with fraction III (6 μg/mL), 6-OHDA (50 μM) and with their combination. Thapsigargine was used as positive control of XBP1 splicing. After 4 h, total RNA of SH-SY5Y cells was extracted. One microgram of DNAse-treated total RNA was retro-transcribed with the Easy-script plus cDNA synthesis Kit (abm) according to manufacturer instructions. Semi-quantitative PCR was performed on 3 μL of cDNA with the following primers: 5′-A AAC AGA GTA GCA GCT CAG ACT GC-3′ and 5′-C CTT CTG GGT AGA CCT CTG GGA G-3′ [55]. The resulted un-spliced and spliced XBP1 mRNA were separated by gel electrophoresis on 3% agarose gel. Then, ethidium bromide-stained amplicons were exposed to LAS 4000 (GE Healthcare, Chicago, IL, USA).

### 2.12. Indirect Immunofluorescence

SH-SY5Y cells were seeded (3 × 10^4^ cells/well) in 12-well plates allowing to adhere for 24 h. Fraction III (6 µg/mL) was pre-incubated for 1 h, then 6-OHDA 50 μM was added for 24 h.

Cells seeded on glass cover slips were washed in PBS, fixed in PBS-4% paraformaldehyde and permeabilized 5 min in PBS containing 0.1% triton. Then, cells were incubated with blocking solution containing PBS-0.5% BSA and 50 mM NH_4_Cl for 30 min [56]. The immunostaining was conducted with rabbit polyclonal anti-α-synuclein (Santa Cruz biotechnology, Dallas, TX, USA). Primary antibody was detected using Alexa fluor 488 antibody (Jackson Immuno Research Laboratories, West Grove, PA, USA).

For mitochondria staining [57], after the treatment, the medium was removed; thus, the cells seeded on glass coverslips were incubated for 30 min at 37 °C in a serum-free medium containing 200 nM Mitotracker Red CMXRos (Invitrogen-Molecular Probes, Waltham, MA, USA). Coverslips were fixed in PBS-4% paraformaldehyde, and then permeabilized in cold acetone for 5 min on ice. After quenching in PBS containing 0.5% BSA and 50 mM NH_4_Cl.

Nuclei were counterstained with 1.6 µM Hoechst 33342 (Sigma Aldrich, St. Louis, MO, USA) for 7 min. Images were acquired on a laser scanning confocal microscope (TCS SP5; Leica MicroSystems) equipped with a plan Apo 63X, NA 1.4 oil immersion objective lens. Quantitative analyses were performed by the ImageJ program, version 1.47 (http://rsb.info.nih.gov/ij/, accessed on 1 February 2022) (N ≥ 30).

### 2.13. Statistical Analysis

Data are reported as mean ± SD of results from three independent experiments. Comparisons between the groups were analyzed by one-way analysis of variance (ANOVA), followed by Bonferroni’s test, with GraphPad Prism 8.0 software (San Diego, CA, USA). Significance was assumed at *p* < 0.05.

## 3. Results

### 3.1. Cocoa Raw Extract Prevents 6-OHDA-Induced Cell Death

First, to evaluate the neuroprotective properties of cocoa in 6-OHDA induced SH-SY5Y cells, we estimated the effect of the cocoa raw extract and, independently, of 6-OHDA on proliferating SH-SY5Y cells (Figure 1). In particular, SH-SY5Y cells were exposed to increasing concentrations of the cocoa extract ranging between 1–50 µg/mL and analyzed by cell viability assay. Our results showed that the cocoa extract induced a substantial reduction of the cell proliferation rate at a concentration higher than 6 µg/mL, when compared to control cells, suggesting that concentrations ranging between 1 and 6 µg/mL were absolutely safe for SH-SY5Y cells (Figure 1A).

The viability of 6-OHDA-treated SH-SY5Y cells was measured using 6-OHDA alone in a concentration range of 5–250 µM (Figure 1B). At this point, the neuroprotective effect of cocoa was assessed by first incubating SH-SY5Y cells with the cocoa raw extract for 1 h, and then by exposing cells to 6-OHDA EC_50_ for 24 h before running the cell viability assay. As a result, the incubation of SH-SY5Y cells with 6-OHDA reduced cell viability to 48.97 ± 2.05% compared to control cells (Figure 1C), whereas pretreatment with of 6 µg/mL of the cocoa extract preserved cell viability up to 67.65 ± 3.14% (*p* < 0.01; Figure 1C).

### 3.2. Cocoa Raw Extract Modulates the UPR Induced by 6-OHDA in SH-SY5Y Cells by Inhibiting Selectively PERK Auto-Phosphorylation

Next, we examined the possible effect of the cocoa raw extract on the unfolded protein response (UPR). Western blots analysis was performed to evaluate the inhibition of PERK phosphorylation and ATF6 cleavage, while RT-PCR experiments were conducted to evaluate the action of the cocoa extract on splicing of XBP1, mediated by IRE1 activation. As shown in Figure 2, the cocoa raw extract was preliminary evaluated for this possible action towards the UPR modulators. The results showed that for ATF6 and IRE1, no modulation was observed, while the induction of PERK phosphorylation with 6-OHDA was inhibited by the cocoa raw extract treatment.

### 3.3. Fractionation of the Cocoa Extract and Identification of Compounds

At this stage, we evaluated the activity on cell viability of single fractions in order to detect the active components of the extract. The phytochemical compounds content in the raw ethanolic extract were revealed by RP-UHPLC-PDA-ESI-MS/MS analysis (Appendix A). The identification of compounds was based on accurate MS and MS/MS spectra, retention time (Rt) of available standards, comparison with literature data [58,59,60,61] and consulting free on-line databases such as ChemSpider (http://www.chemspider.com, accessed on 1 February 2022), SciFinder Scholar (https://scifinder.cas.org, accessed on 1 February 2022) and Phenol-Explorer (www.phenol-explorer.eu, accessed on 1 February 2022).

A bioassay-guided fractionation approach to identify the phytochemicals responsible of biological activity showed by the cocoa raw extract, was applied. In detail, the cocoa extract was purified in three aliquots (Figure 3A): fraction I containing theobromine (1, Rt 1.52 min), the main representative alkaloid found in the cocoa extract (25.59 ± 0.03 mg g^−1^); fraction II composed by catechin-3-O-glucoside (2, Rt 3.07 min) and N-Caffeoyl-L-aspartate (3, Rt 3.46 min); fraction III constituted by epicatechin (5, Rt 4.62 min) and oligomers composed of several unities of catechin and epicatechin such as B-type procyanidin dimer (4, Rt 4.36 min), trimer (6, Rt 6.09 min), tetramers (7, Rt 6.71 min; 8, Rt 6.95 min), pentamer (9, Rt 7.42 min), and hexamer (10, Rt 7.96 min) and A-type procyanidin hexoside (11, Rt 9.01 min) (Figure 3B).

The fraction III represented the only component of the raw extract that guaranteed an increase of viability by the SH-SY5Y cells treated with 6-OHDA. In particular, its administration increased the survival of 24.68 ± 2.87% (*p* < 0.01; Figure 3C).

Based on these results, we tested the single fractions validating the neuroprotective action of fraction III given its selectivity in inhibiting PERK and, consequently, eukaryotic initiation factor 2 (eIF2α) phosphorylation (Figure 3D,E).

### 3.4. Effect of Fraction III on Clonogenic Potential and 3D Spheroids Formation

The ability of fraction III in preserving SH-SY5Y long-term proliferation has been instead assessed by the evaluation of its clonogenic potential and in 3D cell culture systems. A very low subtoxic dose of 6-OHDA (20 μM) significantly reduced the number of colonies and spheroids size formed after 10 days of incubation, whereas the administration of fraction III (6 µg/mL) reverted these effects providing an enhance of clonogenic potential to 53.46 ± 9.23% (Figure 4A) and a spheroids size maintenance comparable to control (Figure 4B).

### 3.5. Fraction III Ameliorates Apoptosis in SH-SY5Y Cells Induced by 6-OHDA

The neuroprotective activity showed by fraction III was also evaluated by flow cytometry through propidium iodide staining. As shown in Figure 5, the incubation of SH-SY5Y cells with 6-OHDA for 24 h augmented the percentage of hypodiploid nuclei, which represent a hallmark of apoptosis. The pre-treatment with 6 µg/mL of fraction III for 1 h reduced the number of cells undergoing apoptosis from 21.65 ± 1.12% (6-OHDA alone) to 8.91 ± 3.67% (fraction III/6-OHDA co-administration) (Figure 5A). Moreover, considering that ER stress response ultimately results in caspase-12-mediated apoptosis [62], we evaluated the inhibition of its activation by the administration of fraction III. A notable reduction in its expression was observed following fraction III administration in 6-OHDA-treated cells (Figure 5B).

### 3.6. Fraction III Attenuates Intracellular ROS and NO Production

Because the exposure of neuronal cells to the 6-OHDA triggers ROS and NO production and taking into consideration the anti-apoptotic proprieties of the fraction III, we investigate the ability of fraction III to reduce the production of such oxidative stress mediators. To this end, SH-SY5Y cells were pre-treated with 6 μg/mL fraction III for 1 h and subsequently exposed to 6-OHDA for 24 h. As a result, we revealed that pre-treatment of SH-SY5Y cells with fraction III reduced both the ROS and the NO released following 6-OHDA induction (Figure 6).

### 3.7. Fraction III Protects SH-SY5Y Cells Induced by 6-OHDA from Mitochondrial Dysfunction and α-Synuclein Accumulation

The continuous accumulations of protein deposits, such as the α-synuclein protein, represents a key signature of several neurological disorders as PD, dementia with Lewy bodies (DLB) and multiple system atrophy (MSA) [63]. Moreover, neurons are highly dependent upon mitochondrial metabolism. Indeed, mitochondrial dysfunction is a hallmark of PD and related forms of parkinsonism.

Considering the PERK autophosphorylation inhibition and antioxidant ability of fraction III, we, then, evaluated the mitochondrial fission and α-synuclein accumulation in the cells. As shown in Figure 7, the treatment of SH-SY5Y cells with 6-OHDA caused a significative increment of α-synuclein protein and mitochondrial fragmentation. Pretreatment with 6 µg/mL fraction III for 1 h considerably reduced α-synuclein build-up and mitochondrial fission, validating the scavenger as well as neuroprotective effect of cocoa.

## 4. Discussion

In this work we investigated the neuroprotective effect of the cocoa extract in a vitro model of PD induced by 6-OHDA, a hydroxylated analogue of the neurotransmitter dopamine, commonly used to mimic PD in animal and cellular models [64]. This compound stimulates a strong neurotoxic effect characterized by an intensive oxidative stress, which increases cellular ROS release [65]. In addition, 6-OHDA can be also accumulated inside mitochondria of dopaminergic neurons and inhibit complex I and III of the electron transfer chain, thus reducing ATP synthesis. Additionally, 6-OHDA generates mitochondrial dysfunction and promotes disease progression in animal models by inducing neuronal cell death in the same way as it occurs in PD patients [66].

Although antioxidant properties have already been recognized for cocoa and derivative compounds, our results revealed for the first time that a cocoa extract is able to protect SH-SY5Y dopaminergic neurons from the neurotoxicity induced by 6-OHDA. Cocoa is known to contain many phytochemicals such as polyphenols, flavanols such as epicatechin and catechin, as well as procyanidins [43]. In fact, several evidence have confirmed that such natural compounds have not only antioxidant but also anti-neuroinflammatory activity and are able to attenuate neurological disorders [54,67,68,69,70]. However, it is reported that cocoa compounds can induce neuronal protection mostly for their efficacy on endothelial cells, in which they stimulate NO synthesis enhancing cerebrovascular circulation [44,45,46]. On the other hand, cocoa-derived products can cross the blood-brain barrier [47] and exert antioxidant activity on neuronal cells (through not yet fully understood mechanisms) [44,48,49].

The present study show that a raw cocoa extract and its purified fraction III prevent cell damage derived from 6-OHDA induced oxidative stress and apoptosis. Similarly, pre-incubation with fraction III clearly shields mitochondrial membranes from the fragmentation produced by 6-OHDA in the neuronal cells. Unbalanced mitochondrial fission is linked to OS and is usually evident in cells derived from PD patients and in experimental models of PD [42,71]. Cocoa protects cells from unbalanced mitochondrial dynamics, which is critical for cell cycle progression and mitochondria turn over via mitophagy [72].

Another important effect of cocoa resides in its protective effect observed on the alteration of α-synuclein expression induced by 6-OHDA in the SH-SY5Y dopaminergic neurons. α-synuclein is a presynaptic protein, which plays important roles in the regulation of synaptic vesicles recycling, particularly, by acting as chaperone for the assembly of the SNAREs proteins, which mediate fusion of vesicles with the synaptic membrane [73]. α-synuclein cycles between two forms: the SNARE bound multimeric form, which localizes at the synaptic membrane, and the cytosolic monomeric form [74]. Under pathological conditions α-synuclein acquires an insoluble β-sheet conformation, whose accumulation starts the early onset of PD and leads to the appearance of cytosolic α-synuclein aggregates of protofibrils, amyloid fibrils and, eventually, Lewy bodies [75,76].

Our experiments clearly show that 6-OHDA induces alteration of the α-synuclein recycling, generating a conspicuous accumulation of the cytosolic form of the protein. However, this event is successfully prevented by pre-incubation of SH-SY5Y with fraction III and, accordingly, our results once again indicate that cocoa derived compounds and those contained in fraction III could prove to be very helpful in the prevention and/or therapy of PD.

We hypothesize that the protective action of cocoa is due to its effect on the UPR. In this regard, we found that pre-incubation with cocoa selectively inhibits PERK autophosphorylation induced by 6-OHDA in SH-SY5Y dopaminergic, thus inhibiting the PERK pathway of the UPR. Moreover, in autoptic tissues of PD patients and in PD experimental models, the PERK pathway of the UPR is found activated and this is believed to be the cause of dopaminergic neuron loss [38,39,42]. Normally, the PERK pathway is activated in response to transient ER stress. PERK autophosphorylation starts an adaptive response characterized by the phosphorylation of eIF2α, which induces not only suppression of protein translation and, consequently, the uptake of potentially unfolded proteins into ER [77,78] but also the preferential mRNA translation of the activating transcription factor 4 (ATF4). This powerful transcription factor activates specific genes, which participate either to the UPR [11,79] and to the antioxidant defense, such as Nrf2 transcription factor, which activates genes involved in the antioxidant response [80]. Conversely, when PERK activation is persistent, as it occurs in PD and other neurodegenerative diseases, the UPR pathway reverts its adaptive response to a proapoptotic response mainly by promoting the expression of the transcriptional regulator C/EBP homologous protein (CHOP) [81]. CHOP, in concert with ATF4, reverts translational repression and can starts the expression of pro-apoptotic factors such as DR5, Trb3, BIM and PUMA [81,82,83,84], which, finally, drive to apoptosis [81,85,86]. Moreover, continual activation of the PERK/CHOP axis enhances ROS production by activating the expression of Ero1 and NOX [87].

For this reason, in PD as well as in other neurodegenerative diseases, the pharmacological or genetic inhibition of the PERK pathway proved to be a successful strategy for protecting cells from OS and apoptosis. Pharmacological inhibition of the PERK pathway can be obtained by specifically targeting PERK autophosphorylation with specific inhibitors. So far, the only specific PERK inhibitor is represented by GSK2606414, which proved to be effective in pink-1 and parkin associated PD models [41] and in a PARK20 early onset PD model [42]. GSK2606414 is a highly selective inhibitor and is also able to cross blood-brain and is considered as promising therapeutic drug for neurodegenerative disorders. Our results clearly show that, similarly to GSK2606414, the cocoa extract and in particular fraction III are able to inhibit PERK phosphorylation and to protect SH-SY5Y cells form all the consequences of 6-OHDA neurotoxicity.

## 5. Conclusions

In a general view, the finding of new therapies that employ natural products is a very attractive goal of biomedical research. On the other hand, the inhibition of the PERK pathway of the UPR is considered an important strategy to be utilized with the aim to attenuate or delay the onset of neurodegenerative diseases and particularly of PD. Nevertheless, few compounds have been evidenced to be effective to inhibit PERK activation and, among them, the GSK2606414 PERK inhibitor is the most widely used [88,89,90]. In the present report, we clearly show that cocoa is able to protect dopaminergic cells from onset of a PD phenotype and that this ability belongs to one or more components of epicatechin-rich and procyanidin-rich cocoa fraction III (Figure 1). Hopefully, further experiments may lead to the identification of novel therapeutic tools to prevent or treat Parkinson’s disease.

## Data Availability

Not applicable.

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
