# Peer review of "Cocoa Extract Provides Protection against 6-OHDA Toxicity in SH-SY5Y Dopaminergic Neurons by Targeting PERK"

_biomedicines, 2022, doi:10.3390/biomedicines10082009_

Round 1

Reviewer 1 Report

Excellent paper. There is very little that may be improved, perhaps some reference to functional protection against 6-OHDA.

certain errata, typographics.

Author Response

  1. “Excellent paper. There is very little that may be improved, perhaps some reference to functional protection against 6-OHDA”.

Reply: We thank the reviewer for this very positive comment. Accordingly, we included ref. 67 and 68 (lane 508 and bibliography section), which highlight the protective role of natural compounds against 6-OHDA toxicity.

Reviewer 2 Report

The authors in this research article determined “Cocoa extract provides protection against 6-OHDA toxicity in SH-SY5Y dopaminergic neurons by targeting PERK”. In the present research article, the authors have assessed the neuroprotective effect of a cocoa extract and its purified fractions against a cellular model of Parkinson's disease represented by 6-hydroxydopamine (6-OHDA)-induced SH-SY5Y human neuroblastoma. Interestingly, the authors revealed the findings for the first time, the ability of cocoa to specifically targets PERK sensor, with significant antioxidant and antiapoptotic activities as crude and fractioning extract. In addition, cocoa also showed antiapoptotic properties in 3D cell model and a notable capacity to block the accumulation of α-synuclein in 6-OHDA-induced cells. Overall, these results indicate that cocoa exerts neuroprotective effects suggesting a novel possible strategy to prevent and mitigate neurodegenerative disorders, such as PD.

I can see very few articles in the present topic, which adds advantage for this study to be novel even though there are some flaws in the study design of the article. I would like to recommend some major concerns to the authors to fulfil the hypothesis, as there are scarce experiments done to show the mechanism in the article.

v  In introduction, the authors need to specify their study aims clearly which can accomplish their hypothesis and which would add some novelty.

v  In materials and methods, the protocols does not show the full details of the experiments, I recommend the authors to include this reference for the above-mentioned protocol, DOI: 10.1016/j.redox.2022.102280

v  I request the authors to include the below studies which have shown significant efficacy of phytochemicals on ER stress mechanism in recent years, please refer this study https://doi.org/10.1016/j.redox.2022.102280, https://doi.org/10.1016/j.apsb.2022.01.017

v  In figure 2 and 3 the authors need to provide the quantification of the blots.

v  In figure 5B the authors need to provide the quantification of the blots.

v  In some of the blots the control blots are alpha tubulin, in some blots GAPDH, why was different controls used, be consistent.

v  In figure 3A, the chromatogram shows several phytochemicals, can the authors show the chemical structures of the phytochemicals in the figure 3A with chromatogram.

v  The authors need to show the final schematic conclusion diagram revealing summarised concluding evidence.

v  A careful English check and grammatical errors need to be resolved by the authors.

The available research information seems to be insufficient, and the authors need to address the above comments. Taking together to all this issue I recommend major revision to the manuscript in present form. 

Author Response

We thank the reviewer for the several suggestions, which we feel will improve the quality of the paper

  1. “In introduction, the authors need to specify their study aims clearly which can accomplish their hypothesis and which would add some novelty”.

Reply: The introduction has been modified as suggested (see lanes 80-87).

  1. “In materials and methods, the protocols do not show the full details of the experiments, I recommend the authors to include this reference for the above-mentioned protocol, DOI: 10.1016/j.redox.2022.102280

Reply: The suggested reference was introduced in the text and in the bibliography section (Ref. 54).

  1.  “I request the authors to include the below studies which have shown significant efficacy of phytochemicals on ER stress mechanism in recent years, please refer this study https://doi.org/10.1016/j.redox.2022.102280, https://doi.org/10.1016/j.apsb.2022.01.017”

Reply: We included the suggested references in the text and in the bibliography section (Ref. 54, 69).

  1. “In figure 2 and 3 the authors need to provide the quantification of the blots”.

Reply: Blots quantifications has been included in figure 1 and 2 as suggested.

  1. “In figure 5B the authors need to provide the quantification of the blots.

Reply: The blots quantifications have been added in figure 5B as rightly suggested.

  1. “In some of the blots the control blots are alpha tubulin, in some blots GAPDH, why was different controls used, be consistent”.

Reply: We apologize about the inconvenience. We did not see differences between the GAPDH and tubuline In Figure 3, the blot was reported with GAPDH just because we ran out of the anti-tubuline antibody.

  1. “In figure 3A, the chromatogram shows several phytochemicals, can the authors show the chemical structures of the phytochemicals in the figure 3A with chromatogram”.

Reply: We modified figure 3A according to the reviewer observations.

  1. “The authors need to show the final schematic conclusion diagram revealing summarised concluding evidence”.

Reply: A scheme (Scheme 1) has been added to summarise conclusions and propose possible mechanism.

  1. “A careful English check and grammatical errors need to be resolved by the authors”.

Reply: We apologize also for this inconvenience. A careful English check has been done on the whole text.

Reviewer 3 Report

Review of manuscript entitled: “Cocoa extract provides protection against 6-OHDA toxicity in SH-SY5Y dopaminergic neurons by targeting PERK” authored by Vincenzo Vestuto, Giuseppina Amodio, Giacomo Pepe, Maria Giovanna Basilicata, Raffaella Belvedere, Enza Napolitano, Alessia Bertamino, Valentina Pagliara, Simona Paladino, Manuela Rodriguez, Alessia Bertamino, Pietro Campiglia, Paolo Remondelli, Ornella Moltedo.

First of all I want to thank for opportunity to review this interesting manuscript.

Neurodegenerative disorders (e.g. Parkinson disease) related with decrease of dopamine in synapses or misfolded protein aggregates are serious and worldwide health problem. In the presented study authors employed neuroblast-like cell line (SH-SY5Y), which is commonly used in studies about Parkinson disease, to investigate possible protective effect of cocoa extract. Authors obtained promising results, which indicate that cocoa extract may exert neuroprotective effects in vitro.

Introduction is very informative, provides many information about the undertaken problem. Methods are described comprehensively. Results are presented clearly. Discussion and conclusions are supported by obtained results.

Overall manuscript is very interesting, well-written, easy to follow and touches important problem.

Major concerns:

·    In the “Statistical analysis” subparagraph, authors stated that “(…)between two groups were analyzed by Student's t-test(…)”, however I see that you have more than two groups (for example: 6-OHDA, III+6-OHDA, Ctrl and III). In my opinion more appropriate approach would be ANOVA (or different test which is capable of comparing more than two groups and is suitable to your research) + post-hoc comparisons.

Minor concerns:

·         In the method sections, please use “x g” instead of “rpm”

Author Response

We thank the reviewer for the important observations he made.

  1. In the method sections, please use “x g” instead of “rpm”

Reply: We made the change as correctly suggested.

  1. In the “Statistical analysis” subparagraph, authors stated that“(…)between two groups were analyzed by Student's t-test(…)”, however I see that you have more than two groups (for example: 6-OHDAIII+6-OHDACtrland III). In my opinion more appropriate approach would be ANOVA (or different test which is capable of comparing more than two groups and is suitable to your research) + post-hoc comparisons.

Reply: We made the change as truly proposed. We adopted the One-way ANOVA with Bonferroni’s test as indicated in the “Statistical analysis” paragraph.

Round 2

Reviewer 2 Report

Most of my previous comments has been answered and the manuscript has been significantly improved. The research article is well written and have discussed the points pertaining their novelty and creates scientific interest for the readers.The available research information seems to be sufficient and advised for publication.